# Motives for Cannabis Use and Risky Decision Making Influence Cannabis Use Trajectories in Teens

**DOI:** 10.3390/brainsci13101405

**Published:** 2023-10-01

**Authors:** Sarah M. Lehman, Erin L. Thompson, Ashley R. Adams, Samuel W. Hawes, Ileana Pacheco-Colón, Karen Granja, Dayana C. Paula, Raul Gonzalez

**Affiliations:** 1Center for Children and Families, Department of Psychology, Florida International University, Miami, FL 33199, USA; erthomps@fiu.edu (E.L.T.); aadam081@fiu.edu (A.R.A.); shawes@fiu.edu (S.W.H.); kgran013@fiu.edu (K.G.); dpaula@fiu.edu (D.C.P.); gonzara@fiu.edu (R.G.); 2Division of Neuropsychology, Department of Neurology, University of Miami Health System, Miami, FL 33136, USA; imp19@miami.edu

**Keywords:** cannabis use motives, decision making, adolescents

## Abstract

The current study will examine the interactive effects of motives for cannabis use (i.e., health or recreational) and risky decision making (DM) on cannabis use trajectories among adolescents. Data from 171 adolescents, aged 14–17 at the initial visit (baseline), were prospectively analyzed across five time points approximately six months apart. Latent growth curve modeling and linear regression analyses were used. We found a significant interactive effect of “recreational motives” and risky DM on the rate of cannabis use over time. Specifically, among those less likely to use cannabis for recreational purposes, riskier DM was associated with a faster increase in the rate of use over time relative to those with lower risky DM. Additionally, a significant main effect showed that those with a greater proclivity to use cannabis for health purposes had higher initial levels of use at baseline and faster increases in the rate of use over time. Regardless of risky DM, using cannabis for health purposes is associated with faster increases in cannabis use escalation. Additionally, risky DM does impact the association between recreational motives for use and cannabis use trajectories. Future work should examine these associations with additional motives for cannabis use that have been previously validated within the literature.

## 1. Introduction

Cannabis is one of the most widely used drugs among adolescents with over one-third of 12th graders reporting use in their lifetime [1]. Relatedly, earlier age of initiation of cannabis use has been associated with a greater risk of cannabis use escalation and the development of a cannabis use disorder [2,3,4]. Though initiation of cannabis use often occurs during adolescence, not all youth will increase their use over time and prior research has found that individual factors play a role in determining which adolescents are at greater risk for escalation of use and the subsequent development of a cannabis use disorder [2,5]. An individual factor that has been associated with cannabis use frequency among adolescents is motives for use [6,7,8]. Similarly, associations have also been found among cognitive factors such as risky decision making (DM) and cannabis use frequency among adolescents [9,10,11]. Given that differences in motives for use and levels of risky DM differentially impact risk of cannabis use escalation among adolescents, the current study seeks to examine the interactive effects of these factors on a sample of adolescent cannabis users who were followed for two years. Examination of these associations may inform prevention and intervention practices by identifying individualized treatments based on an adolescent’s motive for use and level of risky DM.

Within the literature, cannabis use motives demonstrate associations with cannabis use behaviors and consequences of use; however, differences emerge depending on the specific motives that an individual endorses. Ample evidence suggests that using cannabis to cope with negative affect (e.g., using to cheer myself up or to reduce stress) is associated with increased cannabis use frequency and cannabis use-related problems among both adolescents and young adults [12,13,14]. A recent meta-analysis by Bresin and Mekawi [6] analyzed data from 45 cross-sectional studies (*Mage* = 22.35) and found that using cannabis to cope with negative affect showed the strongest association with cannabis use-related negative outcomes (i.e., greater cannabis use frequency and cannabis use-related problems). However, these results are based on prior cross-sectional work, and fewer studies have examined these associations longitudinally. A prospective study examining a nationally representative sample of adolescent cannabis users found that coping with negative affect prospectively predicted increased frequency of cannabis use and use-related problems in emerging adulthood [12]. Similar results have been found with other motives for cannabis use. For example, using cannabis “to get high” and “to be more sociable” tend to be endorsed more frequently than coping motives and have been associated with increased frequency of use among adolescents [12,13,15,16]. However, studies have also found non-significant or negative associations between recreational motives (e.g., using to be more sociable, using to get high, and using to fit in) and frequency of use. A study examining motives for use among a clinical sample of adolescent cannabis users found no significant associations between recreational motives and cannabis use frequency or cannabis use-related problems [17]. Furthermore, a recreational motive that involves using cannabis “to fit in” has been negatively associated with frequency of use and cannabis use-related problems among adolescents. More specifically, a study examining a nationally representative sample of adolescent 12th graders found that using cannabis “to fit in” prospectively predicted less frequent cannabis use at age 35 [15]. Thus, motives may play an important role in predicting differing individual cannabis use trajectories among teens.

Based on the aforementioned research, it appears that using cannabis to cope with negative affect may place adolescents at greater risk for cannabis use escalation. Although prior work has consistently demonstrated an association between mental health motives (i.e., coping with negative affect) and increased frequency of use, far fewer studies have examined this association with physical health motives for use (e.g., using to decrease pain or to sleep better). Given the continued acceptance and legalization of cannabis use for medicinal purposes across the United States, it is important to understand how physical health motives impact frequency of cannabis use. One of the few studies to examine physical health motives for cannabis use was conducted by Lee and colleagues [18] and found that using cannabis to reduce sleep problems was associated with increased frequency of use among first-year college students. Additionally, a more recent study by Chabrol et al. [7] further examined physical health motives for cannabis use by creating a health motives factor (i.e., using cannabis to sleep better, to feel more energetic, to have a better appetite, and to feel healthier) that was based on the authors’ 30-year experience of conducting psychotherapy with adolescent cannabis users. Results from this study demonstrated that the health motives factor was the only significant predictor of both cannabis use frequency and cannabis use-related problems among adolescents. Thus, emerging literature suggests that using cannabis to cope with negative affect and to reduce physical health symptoms is strongly associated with a greater frequency of cannabis use among adolescents. These findings highlight the importance of examining health motives for cannabis use that incorporate both mental and physical health motives as predictors of cannabis use frequency and cannabis use trajectories. The current study seeks to address this gap in the literature by examining a “health motives” factor (i.e., combining mental and physical health motives for use), as well as a “recreational motives” factor as distinct predictors of cannabis use trajectories across adolescence.

Furthermore, given research implicating risky DM in addiction neuropathophysiology [19,20] and evidence suggesting that it may influence cannabis use trajectories among adolescents [11], we also examine its potential moderating role in associations between cannabis use motives and frequency of use. Risky DM involves making a choice between two or more options based on implicit or explicit knowledge of risk [21]. Given that adolescents experience a greater propensity toward risk-taking behavior and are more susceptible to poor DM, it is important to understand how risky DM may influence associations between motives for use and cannabis use trajectories across this developmental period [10,22,23]. For example, we can speculate that adolescents more likely to use cannabis for health purposes (e.g., to reduce pain or anxiety) may increase their use over time at a faster rate if DM is impaired, such that they may have difficulties reducing use if it becomes problematic. However, peers may have more of an influence in cases of recreational use motives (e.g., to have fun with friends), yet DM may also contribute to more problematic use through a variety of mechanisms (e.g., risk taking, disinhibition, and reward sensitivity [19,21,22,24]). Alternatively, it is possible that DM may not influence the association between recreational motives for use and cannabis use trajectories. Prior work has suggested that adolescents often engage in greater risk-taking behavior in the presence of peers [25,26,27]. This may suggest that the influence of peers may be the driving factor for cannabis use escalation among those using for recreational purposes, and that this association may not depend on an individual’s DM ability. Risky DM is an important moderator to consider given its hypothesized role in addiction neuropathophysiology [19,20]. However, this construct has not been examined as a moderator of motives for cannabis use on cannabis use trajectories among teens. 

Additionally, not all DM measures are the same. For example, some measures assess DM under conditions of ambiguous risk (e.g., the Iowa Gambling Task), whereas others assess DM under explicit risk conditions, such as the Game of Dice Task (GDT) [21]. Indeed, a recent study conducted in our lab examined whether specific DM tasks under differing levels of risk (i.e., ambiguous vs. explicit or gain vs. loss), differentially impacted cannabis use escalation among adolescents [11]. Results from this study found that baseline (BL) performance on the GDT significantly predicted greater escalation of cannabis use over time, while no other significant associations were found among other DM tasks (i.e., the Iowa Gambling Task or the Cups Task). Therefore, it appears that measures of explicit risk taking may be more sensitive to problems in DM ability among adolescents, and that the GDT may tap into specific DM constructs (rational-analytic thinking) that identify those at greater risk for cannabis use escalation over time [21,28,29]. It is possible that the latter may become more relevant in emerging adulthood with further development and maturation of the ventromedial prefrontal cortex, which has protracted development and is critical for DM [19,20].

### Aims and Hypotheses

To the best of our knowledge, the current study is the first to examine the interactive effects of motives for use and risky DM on cannabis use trajectories among adolescents. We aim to replicate existing literature on cannabis use motives and frequency of use among a sample of adolescent cannabis users at risk for cannabis use escalation. Furthermore, we hope to extend prior work by exploring whether risky DM moderates any observed associations. We hypothesized that adolescents more likely to use cannabis for health purposes would have higher initial levels of lifetime cannabis use at BL and faster increases in the rate of use over time relative to those less likely to use for health purposes. In contrast, we hypothesized that adolescents more likely to use cannabis for recreational purposes would have lower initial levels of lifetime cannabis use at BL and slower increases in the rate of use over time relative to those less likely to use for recreational purposes. Although we anticipated that risky DM at BL would be associated with faster increases in the rate of cannabis use escalation over time [11], analyses regarding interactions between cannabis use motives and DM were considered exploratory.

## 2. Materials and Methods

### 2.1. Participants and Procedure

Data from 171 adolescent cannabis users (*Mage* = 15.5, *SD* = 0.6) were analyzed as part of a larger longitudinal study (R01 DA031176; PI: RG) that examined associations between DM and cannabis use trajectories [30]. Recruitment took place during the years between 2013 and 2018 throughout the greater Miami area at middle and high schools, parks, movie theaters, and through word of mouth. Screening aimed to identify adolescents who were at risk for cannabis use escalation and whose primary drug of choice was cannabis. However, approximately 10% of the sample enrolled in the larger study reported no history of substance use at BL. By design, these participants were included to protect confidentiality by reducing the risk of participants being labeled as substance users simply through study participation. Eligible participants needed to have the ability to read and write in English and were excluded if they self-reported any history of neurological conditions, birth complications, traumatic brain injury or loss of consciousness greater than 10 min, cannabis or alcohol use disorder, or psychiatric or mood disorders (except for attention-deficit/hyperactivity disorder and conduct disorders, given their high comorbidity with substance use). These criteria served to reduce the presence of potential confounds that have been shown to be associated with cannabis use and cognitive functioning [24,31,32,33]. For more details on inclusion and exclusion criteria see Pacheco-Colón et al. [30]. 

The current study sought to examine associations between motives for cannabis use at BL and cannabis use trajectories among adolescents. Therefore, participants were only included in the current study if they self-reported use of cannabis within six months of their BL assessment and reported on their motives for use at the initial visit. Of the 401 participants enrolled in the larger parent study, 95 did not contribute data because the motives questionnaire was introduced after study onset. Additionally, 112 were excluded from analyses for reporting no cannabis use within the six months prior to the BL assessment, and 36 participants were excluded from analyses for having missing substance use data at one of the five assessment waves. Therefore, a total of 171 participants remained and were included in primary analyses. The sample was majority Hispanic/Latino (91.2%), male (57.3%), and White (73.7%). Further information on participant demographics and characteristics can be found in Table 1. 

Data for the current study were collected at five separate time points scheduled approximately six months apart. The initial visit (BL), one-year follow-up (T3), and two-year follow-up (T5) were completed in person, while the 6-month follow-up (T2), and 18-month follow-up (T4) were completed over the phone. Semi-structured interviews and self-report measures of mental health and substance use were completed at each time point. Neuropsychological testing was administered at three separate time points each one year apart (i.e., BL, T3, T5). Written adolescent assent and parental/guardian consent were obtained prior to each assessment, and the Institutional Review Board at Florida International University provided approval for this study. Additionally, this study was completed in compliance with institutional research standards for human research and in accordance with the Declaration of Helsinki.

### 2.2. Measures

Wide Range Achievement Test 4—Word Reading Subtest (WRAT-4). The WRAT-4 [34] was administered at BL to assess participants’ estimated IQ. The current study used estimated IQ as a covariate in analyses to reduce potential confounds that might influence risky DM and frequency of cannabis use. 

Participant History Questionnaire (PHQ). The PHQ is a semi-structured interview that was administered at BL to assess participant and parental demographic information such as age, biological sex at birth (i.e., male or female), ethnicity, race, and history of psychological disorders. Age at BL and sex were used as covariates in the current study.

Drug Use History Questionnaire (DUHQ). The DUHQ [35,36] is a semi-structured interview used to obtain a detailed self-report history of substance use during a participant’s lifetime, the past six months, and past 30 days. Lifetime frequency of cannabis use (i.e., number of days used) was calculated at BL with past six-month use assessed at subsequent measurement time points. These were summed to obtain a lifetime frequency of cannabis use variable at each assessment wave. Furthermore, lifetime frequency of alcohol and nicotine use were calculated at each time point and used as covariates in analyses. 

Structured Clinical Interview for the Diagnostic and Statistical Manual of Mental Health Disorders, 4th edition (DSM-IV; SCID-IV). The SCID-IV [37] is a semi-structured clinical interview used to assess whether an individual meets criterion for a disorder based on the DSM-IV. The substance use modules from the SCID-IV were used to assess whether participants met criteria for alcohol or cannabis dependence or abuse at each time point. 

Fagerström Test for Nicotine Dependence (FTND). The FTND [38] is a six-item questionnaire used to assess the severity of physical dependence to nicotine. The items are summed to yield a total score ranging from 0–10 with higher scores indicating greater severity of physical dependence. Scores are placed into categories of severity (i.e., 1 = very low dependence, 2 = low dependence, 3 = medium dependence, 4 = high dependence, and 5 = very high dependence). This questionnaire was administered at each time point. 

Marijuana Reasons for Use Questionnaire (MJRUQ). The MJRUQ is a self-report measure of motives for cannabis use developed for the current study. Item selection was based on common motives for recreational cannabis use often reported in the literature [13,15,18,36], as well as symptoms for which cannabis is often used medicinally [13,39,40]. Participants who reported using cannabis within the six months prior to the BL assessment were asked to indicate from a list of 13 items all the motives for why they choose to use cannabis (i.e., they could choose more than one). Items were binary (yes/no). 

For this study, an exploratory factor analysis (EFA) was used to analyze responses from this measure using geomin-rotated loadings. The internal consistency of the measure was found to be adequate (α = 0. 67). Examination of eigenvalues and inspection of scree plots identified two factors with item communalities generally ranging from 0.5 to 0.8. Based on these results, a confirmatory factor analysis (CFA) was conducted to define the latent variables that would be included in our analyses. Results from the CFA showed that the item “to fit in” did not significantly load onto the “recreational” factor and was removed. Following the removal of the item, the CFA was run again. Two latent factors were specified from the CFA with eight items loading onto a “health motives” factor, and four items loading onto a “recreational motives” factor. The fit was fair (χ2 = 551.11, *df* = 66, *p* < 0.01; Root Mean Square Error of Approximation [RMSEA] = 0.06; Comparative Fit Index [CFI] = 0.92; Standardized Root Mean Square Residual [SRMR] = 0.15). Table 2 provides the list of items and factor loadings from the CFA. Both latent factors were then used as predictor variables in subsequent analyses.

Game of Dice Task (GDT). The GDT [41] is a computerized neuropsychological test designed to assess risky DM. Participants were instructed to earn as much fictitious money as possible by guessing the correct number on a single die across 18 trials. Participants chose either a single number or a combination of two, three, or four numbers prior to each trial. Combinations with more numbers (lower risk) were associated with smaller rewards and losses while combinations with fewer numbers (greater risk) were associated with larger rewards and losses. The current study used the number of total risky choices from the GDT at BL as the DM variable of interest. 

### 2.3. Data Analytic Plan

Latent growth curve (LGC) modeling and multiple linear regression analyses were used to examine interactive effects of motives for cannabis use and risky DM on initial levels of lifetime cannabis use at BL (intercept) and rate of lifetime cannabis use escalation over time (slope). All analyses were run in version 8.0 of Mplus [42]. Maximum likelihood parameter estimation (MLR) was used to deal with missing data and standard errors were computed using robust estimators to account for non-normality of the lifetime frequency of cannabis use variable. Estimated IQ, age at BL, lifetime frequency of alcohol and nicotine use, and the GDT predictor variable were grand mean centered prior to running analyses.

An unconditional linear growth model was run to examine average intercept at BL and average slope of lifetime frequency of cannabis use over the two-year study period. We used Hu and Bentler’s [43] recommendations to interpret absolute model fit using the RMSEA, CFI, and SRMR. We then conducted a LGC model with multiple linear regressions to determine whether interactions between motives for use and risky DM predicted the intercept and slope of lifetime frequency of cannabis use across the two-year study period. Sex, estimated IQ, and age at BL were included as time-invariant covariates, and lifetime frequency of alcohol and nicotine use were included as time-varying predictors. The two interaction variables were created in Mplus by multiplying both motives for use latent factors with the risky DM variable (i.e., Health Motives x DM and Recreational Motives x DM). Significant interactions were followed up with simple slopes difference tests [44], with the “health” and “recreational” factors set at one standard deviation above the mean (high “recreational motives” = 1; high “health motives” = 1) and one standard deviation below the mean (low “recreational motives” = −1; low “health motives” = −1). Similarly, the risky DM variable was set at one standard deviation above the mean (high risky DM = 5.15) and one standard deviation below the mean (low risky DM = −5.15).

## 3. Results

### 3.1. Substance Use Characteristics 

Detailed substance use characteristics can be found in Table 3. Lifetime frequency of cannabis, alcohol, and nicotine use generally increased over the five assessment waves. Participants endorsed using cannabis more often than alcohol and nicotine, indicating that enrolled participants endorsed cannabis as their primary drug of choice, as intended. At the final time point, approximately 99% of the sample reported using alcohol in their lifetime and approximately 75% of the sample reported use of nicotine in their lifetime. As the current study included only participants who had endorsed use of cannabis within six months prior to the BL assessment, all participants had reported use of cannabis in their lifetime. Furthermore, the SCID-IV [37] was used to determine which participants met criteria for current (past month) alcohol or cannabis dependence and abuse at each time point. Based on the SCID-IV, 2.9% of the sample met criteria for cannabis dependence and 27.6% met criteria for cannabis abuse at the final assessment wave. For alcohol, no participants met criteria for alcohol dependence, while 2.9% met criteria for alcohol abuse at the final assessment wave. For nicotine, the Fagerström Test for Nicotine Dependence questionnaire [38] was used to assess for physical dependence of nicotine. At the final assessment wave, one participant met criteria for medium physical dependence severity, one participant met criteria for low physical dependence severity, and 36 participants (21.1%) met criteria for very low physical dependence severity. 

### 3.2. The Unconditional Linear Growth Model

The unconditional linear growth model for cannabis use showed adequate fit (RMSEA = 0.22; CFI = 0.95; SRMR = 0.02). Similar to Pacheco-Colón et al. [30], which included both cannabis users and non-users, the average intercept at BL was significant among our cannabis using sample (β = 0.67, SE = 0.05, 95% CI [0.58, 0.77], *p* < 0.001). The average slope was also significant (β = 1.04, SE = 0.06, 95% CI [0.93, 1.16], *p* < 0.001), indicating that lifetime frequency of cannabis use increased over the two-year study period.

### 3.3. The Conditional Latent Growth Curve Model

Within the conditional LGC model, we first analyzed results from the “health motives” latent factor. The interactive effect of “health motives” for cannabis use and risky DM did not significantly predict the intercept or slope; however, we found significant main effects of “health motives” on both the cannabis use intercept and slope (see Table 4). Specifically, at average frequencies of the covariates, those with a greater likelihood of using cannabis for health purposes showed higher initial levels of lifetime frequency of cannabis use at BL and faster increases in the rate of cannabis use escalation over time relative to those less likely to use for health purposes (see Figure 1). 

Results from the “recreational motives” latent factor showed a significant main effect for “recreational motives” on the intercept, such that those who were more likely to use cannabis for recreational purposes had lower initial levels of lifetime frequency of cannabis use at BL, accounting for all covariates (see Table 4). Additionally, a significant interactive effect was found for “recreational motives” and risky DM on the slope (see Table 4). The simple slopes difference test revealed that among those less likely to use cannabis for recreational purposes, higher risky DM was associated with a faster increase in the rate of cannabis use escalation over time relative to those with lower risky DM, whereas among those more likely to use cannabis for recreational purposes, risky DM did not affect rate of cannabis use escalation (see Figure 2). 

Of note, certain covariates were found to be significant predictors of the cannabis use intercept and slope. Specifically, sex as a time-invariant covariate significantly predicted both intercept and slope (see Table 4), such that males had higher initial levels of cannabis use at BL and faster increases in the rate of cannabis use over time compared to females. Additionally, age at BL predicted the intercept such that older participants had higher initial levels of lifetime frequency of cannabis use at BL. Furthermore, lifetime frequency of alcohol use at BL and T2 significantly predicted lifetime frequency of cannabis use at BL and T2, respectively, such that greater frequency of alcohol use was associated with greater frequency of cannabis use. Similarly, lifetime frequency of nicotine use at T4 and T5 significantly predicted lifetime frequency of cannabis use at T4 and T5, respectively, such that greater frequency of nicotine use was associated with greater frequency of cannabis use (see Table 4).

## 4. Discussion

The current study examined the interactive effects of motives for cannabis use and risky DM on cannabis use trajectories among teens. We found support for our hypothesis regarding “health motives”, such that those more likely to use cannabis for health purposes had higher initial levels of use at BL and faster increases in the rate of use over time relative to those less likely to use for health purposes. However, there was no significant interactive effect of “health motives” and risky DM on initial levels of use at BL or the rate of cannabis use escalation over time. This finding suggests that regardless of DM ability, using cannabis for health purposes is a risk factor for faster increases in lifetime cannabis use escalation among adolescents. That is, all youth, including those who perform better on DM tasks, are at risk of cannabis use escalation if they report higher frequencies of health-related reasons for use. In contrast, a significant interactive effect was found for “recreational motives” and risky DM on the rate of lifetime cannabis use escalation over time. Specifically, risky DM did not impact the rate of cannabis use escalation over time among those who were more likely to use for recreational purposes. However, among those less likely to use for recreational purposes, those with riskier DM had a faster increase in the rate of cannabis use escalation over time relative to those with lower risky DM. Indeed, this finding suggests that risky DM does impact the association between motives for use and cannabis use trajectories among adolescents. 

Our findings that “health motives” are associated with greater initial levels of lifetime use at BL and faster increases in the rate of use over time are consistent with prior work showing that using cannabis to reduce negative affect or to reduce physical health symptoms are associated with greater frequency of cannabis use among adolescents [12,13,15]. For example, prior work has found that adolescents using cannabis to relax and relieve tension, to decrease anger/frustration, and to help when feeling depressed have been associated with greater frequency of use [12,13,15]. Fewer studies have examined the physical health motives, which were part of our “health motives” factor; however, similar results have been found. For example, Chabrol and colleagues [7] found that adolescents reporting use of cannabis to sleep better, to feel more energetic, and to have a better appetite were associated with both greater frequency of use and more problematic use symptoms. Additionally, a recent cross-sectional study found that adolescents using cannabis to manage pain, nausea, or another medical problem reported using cannabis more frequently than adolescents using solely for recreational purposes [45]. It is important to consider that approximately one-quarter of our sample endorsed using cannabis to decrease pain or to increase appetite, and that almost half of the sample endorsed using cannabis to sleep better (see Table 2 for additional prevalence rates). Although examined less frequently, these rates along with our results, suggest that physical health symptoms are important factors to consider when examining cannabis use motives among adolescents.

A possible explanation for finding accelerated growth in cannabis use among teens with more health-related motives may stem from the role of both positive and negative reinforcement in addiction [46]. Koob and Le Moal [46] state that positive reinforcement (e.g., using cannabis “to be more social” or “to get high”) initially drives substance use behaviors and that negative reinforcement (e.g., using to decrease anxiety or pain) influences the continuation of substance use behaviors and the development of addiction. We speculate that adolescents who have either a physical or mental health symptom may be engaging in cannabis use for both positive and negative reinforcement, which may exacerbate or accelerate use and development of a use disorder. Given that our motives for use questionnaire allowed participants to endorse as many motives as they would like from the list of 13 items, this would allow them to endorse both “health” and “recreational” motives. Additional research is needed to examine associations between health motives for use and cannabis use trajectories among adolescents who endorse using cannabis exclusively for health purposes.

Partial support was also found for the “recreational motives” for use hypothesis. We hypothesized that those more likely to use cannabis for recreational purposes would have lower initial levels of lifetime use at BL and slower increases in the rate of use over time relative to those less likely to use for recreational purposes. A significant main effect of “recreational motives” on initial levels of lifetime cannabis use at BL revealed that those more likely to use cannabis for recreational purposes (after controlling for health purposes) reported lower initial levels of use at BL. This finding is consistent with prior work examining trends in cannabis use initiation among adolescents. Oftentimes, cannabis use initiation among adolescents is limited to use with peers in social contexts [2]. This might suggest that adolescents in our study who were more likely to use cannabis for recreational purposes (e.g., to be more sociable and to get high) reported lower initial levels of use at BL because they were only using while hanging out with friends. In contrast, adolescents who were less likely to use for recreational purposes might have been more apt to use both with their friends and when they were alone, contributing to the greater frequency of lifetime use found at BL.

Furthermore, unlike the “health motives” factor, we did find a significant interactive effect of “recreational motives” and risky DM on the rate of cannabis use escalation over time after controlling for “health motives”. Results revealed that among those more likely to use cannabis for recreational purposes, risky DM did not impact the rate of cannabis use escalation over time. Rather, we found that among those less likely to use cannabis for recreational purposes, those with riskier DM increased their frequency of use at a faster rate relative to those with less risky DM. One possible explanation for this finding might come from the literature on peer substance use. It is well established within the literature that peer substance use is one of the greatest risk factors for use among adolescents [2]. Therefore, it is possible that friends largely dictate use among individuals who are more likely to use for recreational purposes (e.g., to be more sociable and to have fun), whereas risky DM might play a more important role among individuals who are less likely to use in response to peers.

Another possible explanation for this finding is that an additional cannabis use motive factor might be influencing the results that was not included within the motives for use questionnaire used for the current study. For example, if an individual is less likely to use cannabis for recreational purposes, it might be that they are using for other motives such as “to know myself better”, or “to be more creative” (i.e., expansion motives). Prior work has examined expansion motives as a separate motive factor and has been shown to be associated with an increased frequency of cannabis use among adolescents [6,15]. Although the items used within the MJRUQ for the current study were chosen based on prior literature, future studies should examine whether risky DM interacts with additional cannabis use motives that have been previously validated within the literature [47].

Given that much of the prior work examining motives for use has utilized cross-sectional or prospective analyses with only two time points, the current study adds to the literature by examining growth in cannabis use frequency across multiple time points over a two-year period using LGC modeling approaches, which allows for examination of cannabis use trajectories across adolescence. Additionally, the current study also extended prior work by examining a “health motives” for use factor that incorporated both mental and physical health symptoms. Findings from this “health motives” factor may have important implications for clinical work as adolescents using cannabis to cope with mental or physical health symptoms may be identified as at-risk for cannabis use escalation. Interventions may be able to provide more individualized treatment for adolescents who have a cannabis use disorder by targeting the specific motive that an adolescent endorses. For example, if an adolescent endorses using cannabis to cope with chronic pain, providing appropriate coping skills such as meditation or breathing exercises may decrease the frequency of use [48]. Furthermore, a significant interactive effect was found for recreational motives and risky DM on cannabis use trajectories among adolescents. This finding may also inform interventions by better targeting those most at risk for cannabis use escalation, based on a combination of their reasons for use and their DM performance. 

Despite the longitudinal nature of the current study and other previously mentioned strengths, several limitations should be considered when interpreting our findings. The cannabis use motives measure that was utilized in the current study was developed with two separate cannabis use motives in mind. Although the measure was validated within the current study, future research may use more robust measures for cannabis use motives that include additional motives that were not examined within the current study (e.g., conformity and expansion motives). Additionally, in order to discern the latent factors from the motives measure and use them to test our hypotheses, an EFA and a CFA were conducted using the same sample. Additional studies will need to run these analyses with separate samples in order to further validate the measure. As with all observational studies of substance use history with human subjects, another limitation of our study was the use of self-report measures for substance use as they may be susceptible to recall bias. However, these methods have been used successfully in most observational studies of substance use among human subjects, including several from our team [11,30,36,49]. Furthermore, given that previous work in our lab has found that risky DM as measured by the GDT impacts cannabis use trajectories among adolescents, it was the only measure included in the current study to examine risky DM. Future work may benefit from examining associations between motives for cannabis use and risky DM with a more diverse set of DM measures to determine whether DM under different types of risk (explicit vs. implicit) and reward (loss vs. gain) may influence how motives for use predict cannabis use trajectories among adolescents. Finally, mental and physical health motives for cannabis use were collapsed into one overall “health motives” factor. Although this was suggested by the results of our analyses, future research could expand upon and examine these motives as separate factors in order to examine differences in cannabis use trajectories depending on whether the individual is experiencing physical or mental health symptoms. Despite a sizable sample of 171 participants, more subtle effects may have been missed and we were also precluded from running additional analyses to examine the “health motives” factor as two separate factors. Future studies should examine these associations with larger sample sizes that provide enough statistical power to detect additional significant effects. 

## 5. Conclusions

To the best of our knowledge, this is the first study to examine the interactive effects of motives for cannabis use and risky DM on cannabis use trajectories among adolescents. We found that regardless of risky DM, using cannabis to cope with mental or physical health symptoms was a risk factor for faster increases in the rate of cannabis use escalation over time. This finding has important policy implications as more states continue to legalize medicinal and recreational use of cannabis. Furthermore, risky DM moderated the association between “recreational motives” for use and cannabis use trajectories. Specifically, among those less likely to use cannabis for recreational purposes, those with riskier DM increased their use at a faster rate relative to those with less risky DM. This finding may suggest that an additional cannabis use motive factor might be influencing the results that was not examined within the current study. Future research should attempt to replicate these findings and examine these interactive effects with additional cannabis use motives. 

## Figures and Tables

**Figure 1 brainsci-13-01405-f001:**
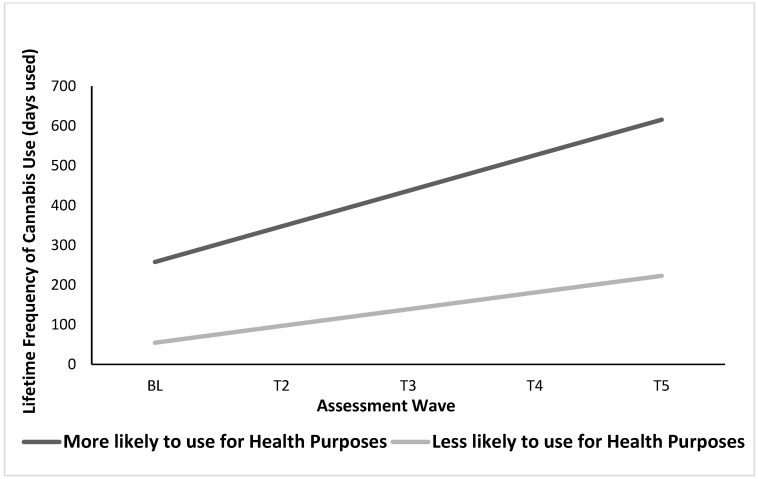
Main Effects of Health Motives for Cannabis Use on Lifetime Frequency of Use over Time. Graph represents cannabis use among males only for easier interpretation.

**Figure 2 brainsci-13-01405-f002:**
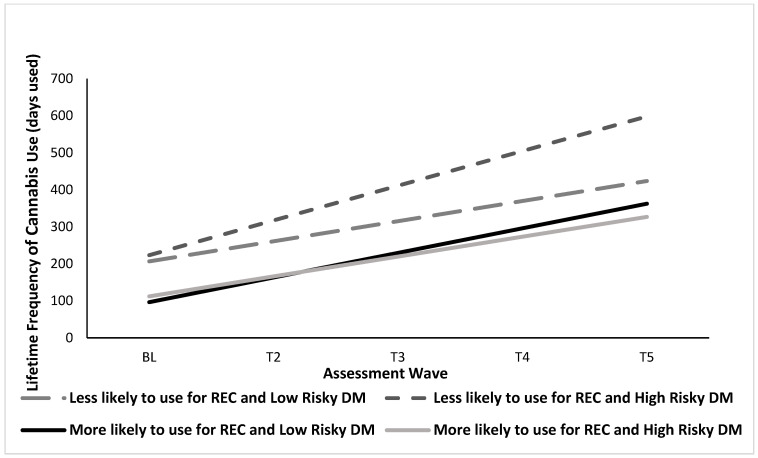
Interactive Effects of Recreational Motives for Cannabis Use and Risky Decision Making on Cannabis Use Frequency over Time. Graph represents cannabis use among males only for easier interpretation.

**Table 1 brainsci-13-01405-t001:** Participant Demographics and Characteristics across Assessment Waves.

(*N* = 171)	BL	T2	T3	T4	T5
Demographics (% or M ± SD)					
% Female	42.7	-	-	-	-
Race and Ethnicity					
% Hispanic White	70.2	-	-	-	-
% Hispanic Black or African American	2.9	-	-	-	-
% Hispanic American Indian/Alaska Native	1.2	-	-	-	-
% Hispanic Native Hawaiian or Other Pacific Islander	0.58	-	-	-	-
% Hispanic More than One Race	12.9	-	-	-	-
% Hispanic Unknown Race	3.5	-	-	-	-
% Non-Hispanic White	3.5	-	-	-	-
% Non-Hispanic Black or African American	4.1	-	-	-	-
% Non-Hispanic More than one Race	1.2	-	-	-	-
WRAT-4 Reading Standard Score	107.7 ± 14.7	-	-	-	-
Years of Education	9.0 ± 0.8	9.8 ± 0.9	10.3 ± 0.8	10.8 ± 0.8	11.3 ± 0.8
Age	15.5 ± 0.6	16.1 ± 0.8	16.5 ± 0.6	17.1 ± 0.7	17.5 ± 0.6
Decision-Making Performance (M ± SD)					
Game of Dice Task (# of risky decisions)	8.3 ± 5.2	-	-	-	-

Note. M = mean; SD = standard deviation; WRAT-4 = Wide Range Achievement Test, Fourth Edition.

**Table 2 brainsci-13-01405-t002:** Confirmatory Factor Analysis Loadings from the Marijuana Reasons for Use Questionnaire.

	Prevalence Rate (%)	Estimate	S.E.	*p*-Value
Health Motives				
To decrease pain	26	0.72	0.06	*p* < 0.001
To sleep better	49	0.77	0.07	*p* < 0.001
To reduce muscle spasms	4	0.61	0.11	*p* < 0.001
To increase appetite	27	0.67	0.08	*p* < 0.001
To reduce nausea	11	0.72	0.08	*p* < 0.001
To reduce anxiety	76	0.94	0.07	*p* < 0.001
To feel happier	57	0.74	0.06	*p* < 0.001
To feel less sad	31	0.58	0.08	*p* < 0.001
Recreational Motives				
To be more social	84	0.39	0.13	*p* < 0.01
To get high	92	0.65	0.12	*p* < 0.001
To have fun	94	0.82	0.13	*p* < 0.001
To be less bored	53	0.78	0.15	*p* < 0.001

Note. Estimates are standardized. S.E. = standard error.

**Table 3 brainsci-13-01405-t003:** Substance Use Characteristics.

(*N* = 171)	BL	T2	T3	T4	T5
Lifetime Days of Use [Md, IQR]					
Alcohol	8 [2, 28]	13 [3, 40]	21 [7, 60]	29 [10, 73]	37 [14, 88]
Nicotine	1 [0, 6]	1 [0, 12]	2 [0, 18]	4 [0, 26]	8 [1, 46]
Cannabis	60 [14, 198]	96 [24, 284]	138 [36, 384]	191 [48, 488]	280 [65, 595]
Lifetime Reports of Substance Use					
% Reporting Alcohol Use in Lifetime	88.9	94.2	97.1	98.2	98.8
% Reporting Nicotine Use in Lifetime	51.5	55.6	64.3	67.8	75.4
Current Substance Dependence					
% Current Alcohol Dependence	0	0	0	0.6	0
% Current Cannabis Dependence	4.1	4.1	4.1	5.3	2.9
Current Substance Abuse					
% Current Alcohol Abuse	0.6	2.9	0.6	0	2.9
% Current Cannabis Abuse	15.2	11.1	22.2	21.1	27.6

Note. Dependence and Abuse criteria are based on the Structured Clinical Interview for the DSM-IV. Md = Median; IQR = Interquartile Range.

**Table 4 brainsci-13-01405-t004:** Results from the Latent Growth Curve Model and Regression Analyses.

	Standardized Estimate	Standard Error	95% CI	*p*-Value
Health → Intercept	0.52	0.08	[0.37, 0.68]	*p* < 0.001 ***
Health → Slope	0.47	0.09	[0.30, 0.64]	*p* < 0.001 ***
Health × DM → Intercept	−0.12	0.09	[−0.29, 0.05]	*p* = 0.16
Health × DM → Slope	0.07	0.08	[−0.09, 0.22]	*p* = 0.39
REC → Intercept	−0.29	0.12	[−0.52, −0.06]	*p =* 0.01 *
REC → Slope	−0.11	0.13	[−0.36, 0.14]	*p* = 0.38
REC × DM → Intercept	−0.01	0.12	[−0.24, 0.24]	*p* = 0.98
REC × DM → Slope	−0.23	0.10	[−0.43, −0.03]	*p* = 0.03 *
DM → Intercept	0.04	0.06	[−0.09, 0.17]	*p* = 0.52
DM → Slope	0.13	0.07	[−0.01, 0.26]	*p* = 0.07
Sex → Intercept	−0.14	0.06	[−0.27, −0.02]	*p* = 0.03 *
Sex → Slope	−0.21	0.07	[−0.35, −0.08]	*p* < 0.01 **
Age → Intercept	0.12	0.05	[0.01, 0.22]	*p* = 0.03 *
Age → Slope	0.03	0.07	[−0.11, 0.18]	*p* = 0.65
WRAT-4 → Intercept	−0.06	0.06	[−0.17, 0.05]	*p* = 0.30
WRAT-4 → Slope	−0.07	0.08	[−0.22, 0.08]	*p* = 0.35
Alc_BL → CU_BL	0.18	0.09	[0.01, 0.36]	*p* = 0.04 *
Nic_BL → CU_BL	−0.03	0.05	[−0.12, 0.06]	*p* = 0.58
Alc_T2 → CU_T2	0.15	0.07	[0.00, 0.29]	*p* = 0.05 *
Nic_T2 → CU_T2	−0.02	0.04	[−0.09, 0.06]	*p* = 0.68
Alc_T3 → CU_T3	0.11	0.06	[−0.01, 0.23]	*p* = 0.06
Nic_T3 → CU_T3	0.02	0.03	[−0.03, 0.07]	*p* = 0.45
Alc_T4 → CU_T4	0.09	0.06	[−0.03, 0.21]	*p* = 0.12
Nic_T4 → CU_T4	0.05	0.02	[0.01, 0.09]	*p* = 0.03 *
Alc_T5 → CU_T5	0.09	0.06	[−0.03, 0.20]	*p* = 0.14
Nic_T5 → CU_T5	0.06	0.03	[0.01, 0.11]	*p* = 0.02 *

Note. Intercept is the initial level of lifetime frequency of cannabis use at baseline. Slope is the rate of cannabis use escalation over the two-year study period. Confidence intervals show the lower and upper limits of the standardized estimate. For sex, 0 = male and 1 = female. CI = confidence interval; Health = health latent factor; REC = recreational latent factor; DM = decision making; REC × DM = interaction between REC latent factor and decision-making variable; Alc = lifetime frequency of alcohol; Nic = lifetime frequency of nicotine; CU = lifetime frequency of cannabis use; BL = baseline; T2 = time point 2; T4 = time point 4; T5 = time point 5. **** p* < 0.001, ** *p* < 0.01, and * *p* < 0.05.

## Data Availability

The data presented in this study are available on request from the corresponding author. The data are not publicly available due to ethical guidelines.

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
