# Peer review of "Motives for Cannabis Use and Risky Decision Making Influence Cannabis Use Trajectories in Teens"

_brainsci, 2023, doi:10.3390/brainsci13101405_

Round 1

Reviewer 1 Report

Dear Authors and Editor,

I thoroughly enjoyed reviewing this article. My congratulations for your work. The methodology is very successful and the instruments have been very well selected. The results are accurate and correct, the tests performed consistent.

Only, I find that it is necessary to visually structure the hypotheses and the objective. These sections should be easy to read and easy to find.

As for the discussion, I liked it very much but I think it lacks more contrast with other studies. I have seen the number of references you have used very scarce, could you add more studies to enrich this section?

Reviewer 2 Report

Thank you for the opportunity to read this text. The article has been very well written. The following are minor suggestions. 

I find it difficult to place this research in time. Maybe it is my inattention, but there should be a starting year (BL) in the abstract and in the main text. It would be good to name it explicitly as a prospective 2-year study with five measurement points, about every six months. Labeling of measurement points varies: in the text and tables T1 to T5, and in figures 0 to 4.

I am against reporting EFA results together with CFAs based on the same sample. This is often even considered an error. I propose to remove the two sentences about EFA, leave internal costintency and full description of CFA.

One might reservations that the description of the results begins with growth models. Since cannabis use is the main dependent variable, I suggest starting with the scale of this phenomenon, perhaps also with a breakdown by gender and by time point. These data are lost in Table 1 included with the sample description. I would pull them into a separate table at the beginning of the results section, along with a commentary.  

Another  concern is about the low sample size, but this is justified by the research procedure with neuropsychological tests. This can be emphasized in the study limitations.

It would also be worth noting here that estimating the number of days of lifetime use of cannabis, alcohol or tobacco may be subject to recall error. Did the authors attempt analyses against a shorter time horizon.

Limitations are often followed by strengths that counterbalance them. The undoubted advantage here is the prospective nature of the research.

Table 1 shows relatively high percentages at risk for addiction. Please explain, how addiction was measured?

It is worth checking the footnotes after table 2. CU = Lifetime frequency of CU sounds strange. Maybe CU = Lifetime frequency of cannabis use. The abbreviation CU did not appear before. It is also better to write WRAT-4 in this table (as earlier, not IQ).

Reviewer 3 Report

This article examined interactive effects of cannabis use motives and risky decision making on adolescent cannabis use trajectories. Using data from a larger longitudinal study and latent growth curve modeling, the authors sought to 1) replicate established findings linking cannabis use motives (recreational and health) with frequency and escalation of use and 2) test risky decision-making as a potential moderator of these associations. This is a clear and generally very well-written manuscript which contributes to literature on both cannabis use motives and risky decision-making. The article would likely be of interest to readers of this special issue, although there are several issues (in particular, lack of detail related to risky decision-making as a moderator) which the authors could address to make the strengthen the manuscript.

1)    The first half of the introduction (covering cannabis use motives) is very thorough, but the paragraphs on risky decision-making were brief and insufficiently detailed. The authors might consider expanding on the rationale for testing risky decision-making as a moderator of the association between motives and frequency/trajectories of use. Why might decision-making tasks assessing different levels or types of risk be differentially impacted by cannabis use escalation?

2)    The authors note that they used performance on the Game of Dice Task (GDT) as a moderator because it has previously been associated with escalation of cannabis use over time. Please be more explicit about why that might be. Are there any relevant features of the GDT that make it the best choice here, aside from the fact that it was associated with use trajectories in a prior study?

3)    A very minor note, but in the final paragraph of the discussion (pg. 3, line 127), the authors write that “...hypotheses regarding interactions between cannabis use motives and DM were considered exploratory.” This should probably read, “...analyses regarding interactions...were considered exploratory.” If not, please include any tentative hypotheses related to the interactions.

4)    Please consider adding a more comprehensive breakdown of race, gender, and ethnicity when presenting demographic data (vs. presenting majority groups, frequently white males, as reference points), even if it is just added to Table 1 and not to the text.

5)    What percentage of participants reported using alcohol and/or nicotine at each time point?

6)    The decision to address health motives instead of just reduction of negative affect was well-motivated and the results are covered thoroughly in the introduction. Have you tried differentiating between physical and mental health motives as IVs in this sample and, if so, was the pattern of results still consistent with what is reported here?

7)    Implications of the findings on recreational and health motives on interventions are addressed. What are potential implications of the results related to risky decision-making as moderator of motives-use associations? Are there any policy-related implications of the use motives and/or decision-making findings?

8)    Regarding future work on risky decision-making, cannabis use motives, and trajectories, are there certain types of decisions (e.g., losses/gains) that you would expect to influence these associations, based on the findings of this manuscript and/or other findings from your group?

Round 2

Reviewer 3 Report

The authors' revisions and responses were very thorough and addressed all of my comments/concerns.